# Associations between Body Mass Index and Prostate Cancer: The Impact on Progression-Free Survival

**DOI:** 10.3390/medicina59020289

**Published:** 2023-02-02

**Authors:** Dorel Popovici, Cristian Stanisav, Marius Pricop, Radu Dragomir, Sorin Saftescu, Daniel Ciurescu

**Affiliations:** 1Department of Oncology, Faculty of Medicine, Victor Babeş University of Medicine and Pharmacy Timisoara, Eftimie Murgu Square 2, 300041 Timisoara, Romania; 2Departments of Radiology, Victor Babeş University of Medicine and Pharmacy Timisoara, 300041 Timisoara, Romania; 3Department of Oral and Maxillo-Facial Surgery, Faculty of Dental Medicine, Victor Babeș University of Medicine and Pharmacy Timisoara, Eftimie Murgu Square 2, 300041 Timisoara, Romania; 4Departments of Obstetrics and Gynecology, Victor Babeş University of Medicine and Pharmacy Timisoara, 300041 Timisoara, Romania; 5Department of Medical and Surgical Specialties, Faculty of Medicine, Transylvania University of Braşov, 500019 Braşov, Romania

**Keywords:** prostate cancer, obesity, overweight, body mass index, age, metastases, PSA

## Abstract

*Background and objectives:* This study aimed to evaluate the impact of body mass index on PCa outcomes in our institution and also to find if there are statistically significant differences between the variables. *Materials and Methods:* A retrospective chart review was performed to extract information about all male patients with prostate cancer between 1 February 2015, and 25 October 2022, and with information about age, weight, height, follow-up, and PSA. We identified a group of 728 patients, of which a total of 219 patients resulted after the inclusion and exclusion criteria were applied. The primary endpoint was progression-free survival, which was defined as the length of time that the patient lives with the disease, but no relapses occur, and this group included 105 patients. In this case, 114 patients had a biological, local or metastatic relapse and were included in the progression group. *Results:* Our study suggests that prostate cancer incidence rises with age (72 ± 7.81 years) in men with a normal BMI, but the diagnostic age tends to drop in those with higher BMIs, i.e., overweight, and obese in the age range of 69.47 ± 6.31 years, respectively, 69.1 ± 7.51 years. A statistically significant difference was observed in the progression group of de novo metastases versus the absent metastases group at diagnostic (*p* = 0.04). The progression group with metastases present (n = 70) at diagnostic had a shorter time to progression, compared to the absent metastases group (n = 44), 18.04 ± 11.37 months, respectively, 23.95 ± 16.39 months. Also, PSA levels tend to diminish with increasing BMI classification, but no statistically significant difference was observed. *Conclusions:* The median diagnostic age decreases with increasing BMI category. Overweight and obese patients are more likely to have an advanced or metastatic prostate cancer at diagnosis. The progression group with metastatic disease at diagnostic had a shorter time to progression, compared to the absent metastases group. Regarding prostate serum antigen, the levels tend to become lower in the higher BMI groups, possibly leading to a late diagnosis.

## 1. Introduction

Globally, according to a study in 2016, 39% of adults aged 18 years old and older were overweight (39% of men and 40% of women) and about 13% obese (11% of men and 15% of women) [1]. Obesity is a major public concern around the world because it is also a disease in its own right and is now considered to be a cause of at least 13 types of cancer [2]. Each year, it is estimated that 19 million new cancer cases are diagnosed worldwide—around 10 million cases are in men and 9.2 million in women, with a mortality rate of 54%, and 43%, respectively [3]. The biological pathway between obesity and cancer is currently being investigated, but not completely understood. Sakers et al. suggest that the negative health effects of obesity come from physiologic stimuli that induce alterations in adipose tissue metabolism, structure, and phenotype. Simply explained, adipocytes lose their plasticity causing a diminished or aberrant response to signaling and promoting the pathological outcome [4]. Obese people suffer from significant metabolic and endocrinological abnormalities that lead to enhanced insulin and insulin-growth factor signaling, dysregulation of sex hormone metabolism, and adipose tissue-derived inflammation [2,5]. Experimental animal models have shown that obesity leads to cancers of the mammary gland, colon, skin, and prostate [6].

According to Globocan, in 2020, prostate cancer (PCa) was among the most diagnosed cancers worldwide, with around 1.4 million men affected by the disease (7.8%), being surpassed by breast (2.2 million), lung (2.2 million) and colorectal cancer (1.9 million). Globally, prostate cancer was the second most diagnosed cancer in male subjects (15.1%), following lung cancer (15.4%), and taking the fifth place in the mortality rate [3].

In Europe, the estimated number of new cases of PCa was ~470,000 (20% of the male total), which makes it the most frequent cancer diagnosed in men. Additionally, the cumulative risk of being diagnosed with prostate cancer before the age of 75 is 8.2% (1 in 12 men), while the risk of PCa death before the age of 75 is 1% (1 in 103 men) [7].

In Romania, 8055 new PCa cases were diagnosed in 2020, representing 8.15% among all cancers in men aged 45+, and taking second place, after lung cancer [3,7]. The proportion of men diagnosed with PCa before the age of 60 is 1.2%, with a mortality rate of 0.2%, and after the age of 60 is 14.2%, with a mortality rate of 7.5% [3].

Studies suggest that nonmodifiable risk factors, besides age, include several others such as family history of cancer, height, lower testosterone level, type 2 diabetes, higher serum glucose, and high insulin levels. Accounted as significant modifiable risk factors are overweight and obesity, high intake of red meat, fat, dairy, and eggs, consumption of fish, and soy foods, tobacco smoking, and alcohol consumption [8].

Over time, PCa incidence and mortality were significantly different during the past years, worldwide, and they seem tightly correlated to the use of prostate-specific antigen (PSA) measurement in the male population [9]. Incidence rates for PCa are estimated to rise by +71.6% worldwide, followed by a rise of +97.1% in mortality rate, between 2020 and 2040. The highest incidence will be registered in Africa (+106.8%), Asia (+94.1%), Latin America and the Caribbean, (+81.5%), and Oceania (+47.7%), followed by the lowest incidence rates in Europe (+27.6%) and Northern America (+23.5). The mortality rate will also rise significantly on all continents, with Asia (+112.7%) leading the charts, followed by Africa (112.3%), Latin America and Caribbean (+110.4%), Oceania (+92.5%), Northern America (+80.7%) and Europe (+53.2%). In Romania, the incidence is estimated to rise by +21.5%, followed by a mortality rate of +31.2% [3].

The current study aimed to evaluate the impact of body mass index (BMI) on PCa outcomes in our institution and also to find if there are statistically significant differences between the variables.

## 2. Materials and Methods

### 2.1. Criteria

We performed a retrospective chart review to extract information about all male patients with prostate cancer seen in our electronic health record system between 1 February 2015 and 25 October 2022, with information about age, weight, height, follow-up, and PSA. We identified 728 patients, of which we excluded 509 patients due to missing information on height, weight; PSA measured in other medical laboratories; low body mass index; association of other malignancies and patients missing from follow-up visits. This resulted in a total of 219 patients who were included in the final analysis. The primary endpoint was progression-free survival (PFS), which was defined as the length of time that the patient lives with the disease, but no relapses occur, and this group included 105 patients. In this case, 114 patients had a biological, local or metastatic relapse and were included in the progression group (Figure 1). The body mass index (BMI) of each patient was calculated using weight and height, documented in the patient medical history. This analysis is partly based on self-declaration of weight and height, which might be underestimated by the patients, which might lead to potential deviations. Data about the radical prostatectomy, orchiectomy, lymphadenectomy and pathological staging is limited because the surgeries and the histopathological exam were performed in other hospitals. This study was approved by the committee board members of OncoHelp Association Timisoara.

### 2.2. Statistical Analysis

Numeric variables were expressed as mean (±SD) and discrete outcomes as absolute and relative (%) frequencies. We created three groups according to the values of BMI. Group comparability was assessed by comparing baseline follow-up duration between groups. Normality and heteroskedasticity of continuous data were assessed with Shapiro-Wilk and Levene’s test, respectively. Continuous outcomes were compared with ANOVA, Welch ANOVA, or Kruskal-Wallis tests according to data distribution. Discrete outcomes were compared with chi-squared or Fisher’s exact test accordingly. The alpha risk was set to 5% and two-tailed tests were used. The difference between ages according to modalities of BMI was assessed with the ANOVA. 

If the null hypothesis of the ANOVA test was rejected, post-hoc pairwise analyses were performed with Tukey’s HSD test. The alpha risk was set to 5% (α = 0.05). Statistical analysis was performed with EasyMedStat—version 3.20; www.easymedstat.com (accessed on 21 October 2022).

## 3. Results

### 3.1. Patient Population

The clinical characteristics of patients are shown in Table 1. Of the 219 patients, 28% (62) were categorized as normal weight (NW) (18.5–24.9 kg/m^2^), 43% (94) as overweight (OW) (25–29.9 kg/m^2^), and 29% (63) as obese (OB) (30+ kg/m^2^).

### 3.2. BMI and Age

We identified a statistically significant difference (*p* < 0.05) regarding age in the BMI categorized groups. The median age for the NW group was 72 ± 7.81 years vs. 69.47 ± 6.31 years in the OW group and 69.1 ± 7.51 years in the OB group (Table 1, Figure 2). The difference between ages according to modalities of BMI was assessed with ANOVA. The null hypothesis was rejected (*p* = 0.007), so we performed a post-hoc Tukey test to explore the differences between the means of all three groups and observed a significant difference between the NW and OW groups (*p* = 0.023) and NW and OB groups (*p* = 0.0087; Table 2). There was no significant difference (*p* = 0.9) between the OW and OB groups.

### 3.3. BMI and cTNM Stage

Group stages in correlation with BMI were represented as follows: stage 2–4.84% NW, 11.70% OW, and 14.28% OB: stage 3–30.65% NW, 31.91% OW and 23.81% OB and stage 4–64.52% NW, 56.39% OW and 61.90%. No statistically significant difference was found between the groups (*p* = 0.567).

### 3.4. BMI and De Novo Metastases

De novo metastasis rates were 51.61%, 46.81%, and 46.03% in patients categorized as NW, OW, and OB, respectively. No statistically significant difference was found (*p* = 0.788).

### 3.5. BMI and Recurrent Metastases

Recurrent metastasis rates were 19.35%, 25.53%, and 20.63% in patients for which BMI was NW, OW and OB. No statistically significant difference was found between the groups (*p* = 0.614).

### 3.6. BMI and Gleason Score

The Gleason score ≥4 + 3 = 7 was most seen between the groups, with 46 (72.19%) in the NW group, 71 (74.50%) in the OW group and 49 (77.77%) in the OB group. No statistically significant difference was found between the groups, overall (*p* = 0.905).

### 3.7. BMI and GnRH Agonists

Leuprorelin and Triptorelin were the most used agonists among PCa patients (rates of 58.06%, 59.57%, and 73.02% in the NW, OW and OB groups). No statistically significant difference was found between GnRH agonists (*p* = 0.147).

### 3.8. BMI and Bisphosphonates

The use rates of zoledronic acid were 20.96%, 10.64%, and 11.11% within the NW, OW, and OB groups, respectively, with no statistically significant differences (*p* = 0.422).

### 3.9. BMI and Radical Prostatectomy or Orchiectomy

Approximately ¼ of the patients from each BMI category group underwent radical prostatectomy (24.19%, 26.61%, and 23.81% for the NW, OW, and OB groups), while orchiectomy was applied in much lower rates (3.23%, 4.32%, and 5.0), with no statistically significant difference between therapeutic approaches within the groups (*p* = 0.907 and *p* = 0.914).

### 3.10. BMI and PSA

The median PSA for the NW group was 123.45 ± 366.58 vs. 48.67 ± 132.6 in the OW group and 54.23 ± 156.6 in the OB group, with no statistically significant difference (*p* = 0.1).

### 3.11. Progression Group According to De Novo Metastases

Overall, 114 events that define progression of PCa were included in this analysis. 70 patients (61%) in the group of events were recorded with de novo metastases and 44 (39%) were without. The median progression time for those patients who presented with de novo metastasis was 15.91 months, while 20.17 months was for those without de novo metastases. The Mann-Whitney test was used to compare the progression group median according to de novo metastasis. There was a statistically significant difference in the progression group between the patients with present and absent metastases (*p* = 0.04, Table 3).

### 3.12. Progression Group According to Recurrent Metastases

From the same group of 114 patients that experienced events, 49 patients (43%) in the group presented with recurrent metastases and 65 patients (57%) did not have recurrent metastases. The median progression time was 16.8 and 17.03 months for patients with, and without recurrent metastasis, respectively. The Mann-Whitney test was used to compare the median progression time according to recurrent metastasis. There was no statistically significant difference (*p* = 0.859, Table 4).

### 3.13. Progression Group According to BMI

Finally, from the same group of 114 patients who presented with progression of the disease, 32 (28%) were NW, 46 (31%) were OW, and 36 (41%) were OB. The median time of progression was 15.42, 19.56, and 17.01 months for the patients categorized as NW, OW, and OB, respectively. The Kruskal-Wallis one-way analysis of variance was used to compare the median time in the progression group according to BMI. There was no statistically significant difference between NW and the other two groups (*p* = 0.4, Table 5). We also performed a Mann-Whitney test to compare in the progression group the median time between NW and OW, NW and OB, and OW and OB groups. There was no statistically difference between NW and OW (*p* = 0.3), NW and OB (*p* = 0.1), and OW and OB (*p* = 0.2).

We used the Kaplan-Meier method to estimate the progression of PCa as an end-point, from the diagnostic date until the date of the last consultation in the progression group. The log-rank non-parametric test for comparison of progression distributions was used to compare recurrence differences between the NW, OW, and OB groups. There was no difference between the recurrence distributions (*p* = 0.09, Figure 3). 

## 4. Discussions

Prostate cancer remains a global health burden and cases will continue to rise, but pharmacological and technological advances are considerably improving and should enable future precision and improved clinical outcomes [10]. A proof-of-concept for future precision was recently demonstrated in some studies, where the teams used a Transient Receptor Potential Melastatin 8 channel (TRPM8) agonist, an ion channel in the plasma membrane, encapsulated into a Lipid NanoCapsule in order to inhibit PCa cell migration. The use of TRPM8 has a great impact in castration-resistant prostate cancer, which is considered to be the most aggressive form of PCa, because it is usually resistant to androgen deprivation therapy [11,12]. However, it is clear that over time a high index of body mass will influence our general physical well-being, along with age and other factors, and can lead to negative results in cases of pathologies, such as cancers.

Our study suggests that prostate cancer incidence rises with age (72 ± 7.81 years) in men with a normal BMI, but the diagnostic age tends to drop in those with higher BMIs, i.e., overweight, and obese in the age range of 69.47 ± 6.31 years, respectively, 69.1 ± 7.51 years. A single-center retrospective study from 2022 investigated the relationship between BMI and prostate cancer risk in 1079 Italian men, which also revealed that overweight and obese men were diagnosed at younger ages, compared to normal-weight patients. Furthermore, excessive fat accumulation contributes to a more favorable tumor microenvironment onset and growth [13].

We also observed, as well mentioned in the study cited above, that PSA tends to diminish with increasing BMI classification, but no statistically significant difference was observed. Both factors, higher BMI and lower PSA, can lead to a high-grade PCa, avoiding an early diagnostic, also seen in this study, where almost half of OW and OB patients had a stage IV PCa diagnostic.

A study from 2017 suggests that long-term weight gain is associated with an increased risk of high-grade PCa among never smokers and among men who were overweight or obese at age 21 [14]. Furthermore, OB men tend to have higher levels of insulin, insulin-like growth factor-1, and lower level of androgens and adiponectin, suggesting that inflammatory and hormonal pathways are involved. Prolonged hyperinsulinemia raises the bioavailability of IGF-1, which has been shown to promote proliferation and inhibit apoptosis in normal prostate and tumor cells in vitro, increasing the risk of PCa [14,15,16,17,18]. Adiponectin is secreted by the adipose tissue and is inversely related to the degree of adiposity [19]. This protein hormone promotes apoptosis and inhibits proliferation and angiogenesis, and higher concentrations have been shown to decrease the risk of high-grade prostate cancer [14,19] Opposed to adiponectin, leptin concentrations are directly related to adiposity and the biological effects are to stimulate cell proliferation and promote angiogenesis [20].

The immune response could also play an important role in the progression of PCa. In a recent study by Fujita et al., they researched the relationship between high-fat diet (HFD) induced inflammation and tumor progression PCa in mice and found that local inflammation of the prostate is one of the most important factors for the progression of PCa in obese and HFD-fed mice in early and late stages. Interestingly, the number of B cells, T cells, macrophages, and mast cells and the ratio of CD8/CD4 T cells were not changed by the HFD, but the number of myeloid-derived suppressor cells and the M2/M1 macrophage ratio were significantly increased in the HFD-fed mice compared with the control group. Using celecoxib, a cyclooxygenase 2 inhibitor, the promotion of tumor growth by the HFD was canceled, which suggests that inflammation plays a specific role in tumor progression caused by HFD. All these hormonal and immune changes in obese people lead to chronic inflammation and play an important role in the development and progression of PCa [21].

We observed a statistically significant difference in the progression group of de novo metastases versus the absent metastases group at diagnostic (*p* = 0.04). The progression group with metastases present (n = 70) at diagnostic had a shorter progression time, compared to the absent metastases group (n = 44), 18.04 ± 11.37 months, respectively, 23.95 ± 16.39 months. No other statistically significant difference was found between the progression and recurrent metastasis group (*p* = 0.859) or BMI (*p* = 0.4).

Our study has its limitations. Since the cohort study has not had enough participants, it is possible that the results could not meet the criteria for significance. For example, the progression of PCa in the BMI groups was not statistically significant between the NW group and the OW, or OB group, but looking at the Kaplan-Meier curve, the NW line has a more obvious decrease at some point, compared to the OW and OB groups. Some studies suggest that there is an inverse association in the progression of the disease between obese and normal-weight patients, where normal-weight patients have a faster relapse and obesity has a protective effect from the PCa recurrence [22,23]. This inverse association is called the obesity paradox. A study conducted by Martini et al. and another study by Schiffmann et al., revealed that obese patients treated with docetaxel and prednisone for metastatic castration-resistant prostate cancer benefited of a protective factor against overall mortality and death, respectively, the second team mentioned that increased BMI was associated with a decreased risk of metastases after radical prostatectomy [24,25]. We propose that urologists should be attentive to radical prostatectomy procedures in overweight and obese patients in order to avoid positive surgical margins. Marenco et al. suggests the fluorescent confocal microscopy as a novel technique that could be used for real-time diagnosis of PCa and also for the evaluation of surgical margins during radical prostatectomy. An advantage of fluorescent confocal microscopy, compared to other intraoperative histological evaluations, could be the rapid application to whole tissue sections [26]. Regarding lymph node dissection, we think that a more aggressive approach is currently suited for the patients with increased BMI, but we believe that further studies need to be corelated with the molecular mechanisms underlying PCa migration, in order to enable a better clinical, surgical and pharmacological management. Such molecular mechanisms were studied recently in vivo and revealed that the upregulation of HGK (a component of Mitogen-Activated Protein Kinase Kinase Kinase Kinase 4), Culin 4B (a scaffold protein with oncogenic activity), overexpression of Human Homebox B9 (a key transcription factor that promotes metastases) and low levels of Receptor tyrosine kinase-like receptor (a noncanonical Wnt receptor) play a major role in the aggressive behavior of PCa [27,28,29,30]. Our data about the radical prostatectomy, orchiectomy, lymphadenectomy and pathological staging is limited because the surgeries and the histopathological exam were performed in other hospitals.

## 5. Conclusions

Our results suggest that the median diagnostic age decreases with increasing BMI category. Furthermore, overweight, and obese patients are more likely to have an advanced or metastatic prostate cancer at diagnosis. In the metastatic group we observed that the progression of the disease has a shorter interval, leading to a faster relapse. The levels of prostate serum antigen tend to become lower in the higher BMI groups, possibly leading to a late diagnosis. Further studies are needed to determine if there is an inverse association between progression of prostate cancer in normal weight and overweight or obese patients, involving excessive fat as a protective mechanism against prostate cancer. We want to address a further question. Does the subcutaneous fat exert a protective effect on the development and recurrence of prostate cancer or other pathways are involved?

## Figures and Tables

**Figure 1 medicina-59-00289-f001:**
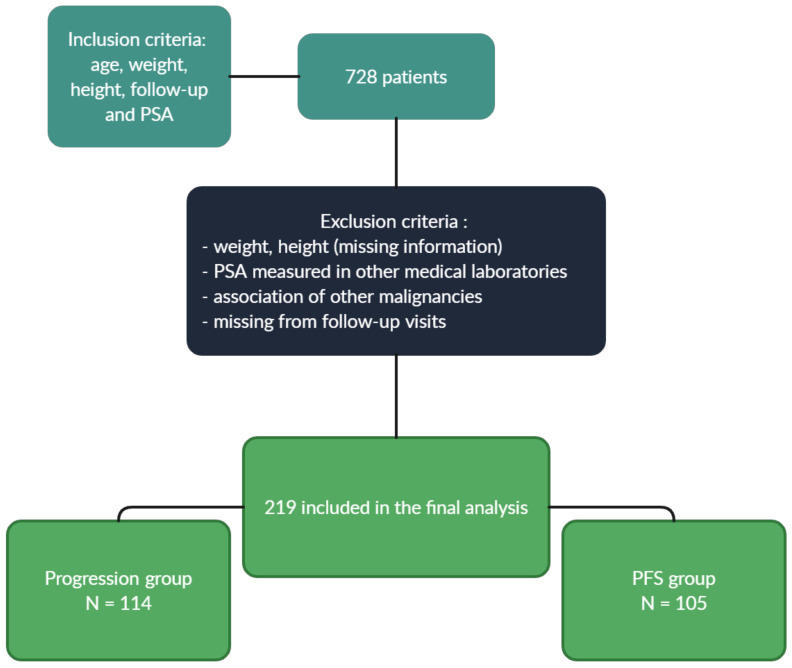
Inclusion and exclusion criteria.

**Figure 2 medicina-59-00289-f002:**
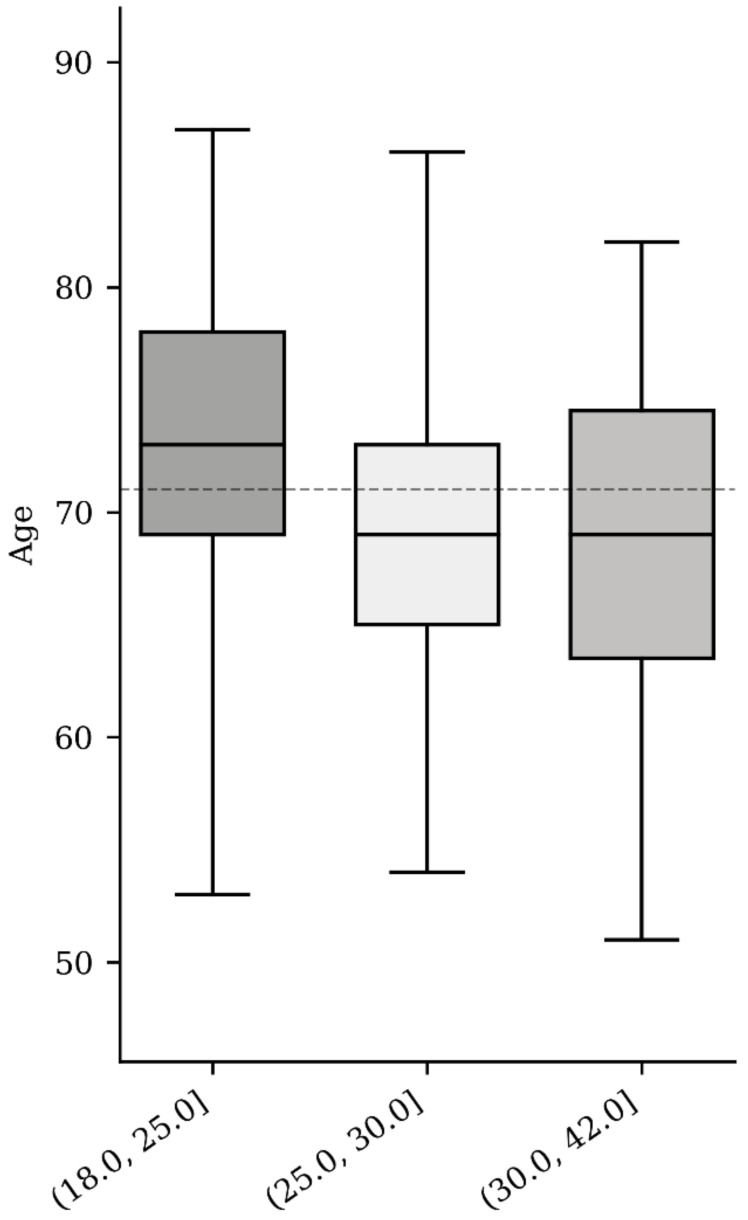
BMI based on age.

**Figure 3 medicina-59-00289-f003:**
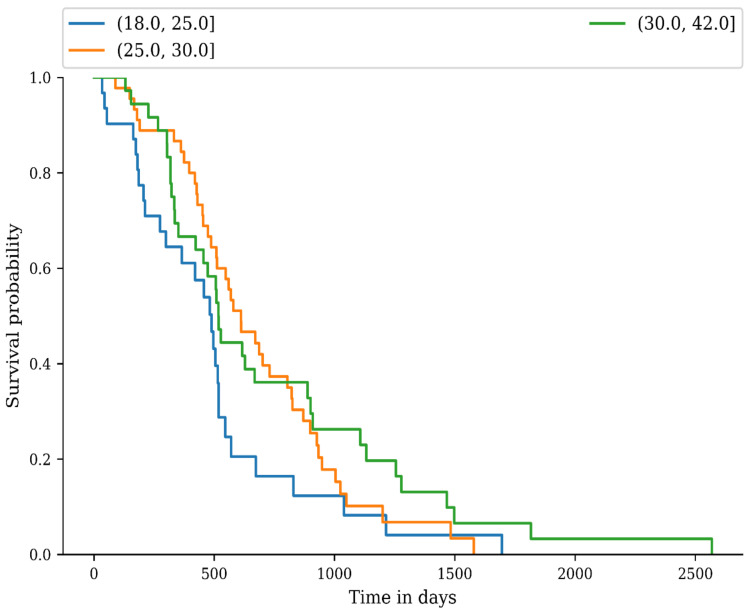
Progression of prostate cancer based on BMI.

**Table 1 medicina-59-00289-t001:** Patient clinical characteristics.

Variable	(18.0–24.9) N = 62 (28%)	(25.0–29.9) N = 94 (43%)	(30.0–42.0) N = 63 (29%)	*p*-Value
**Age**	72.68 (± 7.81) 95% CI: [70.69; 74.66] Range: (53.0; 87.0)	69.47 (± 6.31) 95% CI: [68.18; 70.76] Range: (54.0; 86.0) RR 1.42 OR 2.44	69.1 (± 7.51) 95% CI: [67.2; 70.99] Range: (51.0; 82.0) RR 1.48 OR 2.63	0.007
**cTNM Stage**				
**2a** **2b** **3a** **3b** **3c** **4a** **4b**	0 (0.0%) 3 (4.84%) 6 (9.68%) 13 (20.97%) 0 (0.0%) 9 (14.52%) 31 (50.0%)	6 (6.38%) 5 (5.32%) 8 (8.51%) 19 (20.21%) 3 (3.19%) 12 (12.77%) 41 (43.62%)	3 (4.76%) 6 (9.52%) 3 (4.76%) 11 (17.46%) 1 (1.59%) 10 (15.87%) 29 (46.03%)	0.567
**Metastases** **De novo**	32 (51.61%)	44 (46.81%)	29 (46.03%)	0.788
**Recurrent**	12 (19.35%)	24 (25.53%)	13 (20.63%)	0.614
**Gleason score**				
**≤3 + 4=7** **>4 + 3=7**	15 (27.80%) 46 (72.19%)	24 (25.50%) 71 (74.50%)	14 (22.22%) 49 (77.77%)	0.905
**GnRH agonist**	36 (58.06%)	56 (59.57%)	46 (73.02%)	0.147
**Bisphosphonates**	13 (20.96%)	10 (10.64%)	7 (11.11%)	0.422
**Radical prostatectomy**	15 (24.19%)	25 (26.61%)	15 (23.81%)	0.907
**Orchiectomy**	2 (3.23%)	4 (4.32%)	3 (5.0%)	0.914

GnRH—gonadotropin-releasing hormone.

**Table 2 medicina-59-00289-t002:** Tukey HSD test results.

Pairwise Comparisons		HSD_0.05_ = 2.84	Q_0.5_ = 3.34
		HSD_0.01_ = 3.54	Q_0.5_ = 4.12
**NW–OW**	G_1_ = 72.69	3.19	Q = 3.76 (*p* = 0.023)
	G_2_ = 69.49		
**NW–OB**	G_1_ = 72.69	3.59	Q = 4.23 (*p* = 0.0087)
	G_3_ = 69.10		
**OW–OB**	G_2_ = 69.49	0.4	Q = 0.47 (*p* = 0.94)
	G_3_ = 69.10		

NW—normal weight; OW—overweight; OB—obese; HSD—honestly significant difference.

**Table 3 medicina-59-00289-t003:** Progression group and de novo metastasis.

De Novo Metastasis	Present	Absent
Mean ± SD	18.04 ± 11.37 (months)	23.95 ± 16.39 (months)
Median	15.91	20.17
Min–Max	0.0–55.76	1.12–84.43
N	70	44

**Table 4 medicina-59-00289-t004:** Progression group and recurrent metastasis.

Recurrent Metastasis	Present	Absent
Mean ± SD	20.51 ± 13.17 (months)	20.18 ± 14.3 (months)
Median	16.8	17.03
Min–Max	1.12–59.7	0.0–84.43
N	49	65

**Table 5 medicina-59-00289-t005:** Progression group and body mass index.

BMI	(18.5–24.9 kg/m^2^)	(25.0–29.9 kg/m^2^)	(30.0–42.0 kg/m^2^)
Mean ± SD	15.76 ± 11.64 (months)	21.24 ± 11.36 (months)	23.22 ± 17.23 (months)
Median	15.42	19.56	17.01
Min–Max	1.12–55.76	0.0–51.91	4.31–84.43
N	32	46	36

## Data Availability

The data generated or analyzed during this study are included in this published article or are available from the corresponding author on reasonable request.

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
