# Peer review of "Associations between Body Mass Index and Prostate Cancer: The Impact on Progression-Free Survival"

_medicina, 2023, doi:10.3390/medicina59020289_

Round 1
Reviewer 1 Report
In this study the authors sought to clarify the role of BMI in PCa patients.
Please follow these point for major revision.
Methods
- It's not clear what was the target population. Metastatic de nove vs recurrent PCa after treatment with radical/curative intent. Please add a Consort Flow Diagram to graphically describe the inclusion and the reasons for exclusion of the all sample.
Results
- 55 pts received Radical Prostatecomy as primary - curative - treatment. Please add the pathological stage.
- How many of those received a lymphadenectomy? What was the yield of pelvic lymph node dissection?
- Please add the cTNM stages.
If these information are not available please add as limitation.
Discussion
- It must be discussed the obesity paradox among patients with metastatic castration-resistant prostate cancer (mCRPCa - doi: 0.1038/s41391-021-00418-0) and after radical surgery (10.1007/s00345-018-2240-8). Please further discuss these papers in an appropirate section.
-Hence, what was the proposal of the authors - for example - in intention-to-cure setting (Radical Prostatectomy- Radical RT)? A more aggressive surgical management in order to perform a debulking procedures in a well selected cohort? Increasing the template of the lymph node dissection for example with intraoperative guidance (doi: 10.2967/jnumed.120.259788; doi: doi: 10.1111/iju.14513)? Avoidance of positive surgical margins using intraoperative confocal microscopy (doi: 10.1016/j.euf.2020.08.013; doi: 10.1016/j.eururo.2021.03.021). The authors should consider to discuss these evidence.
Reviewer 2 Report
The manuscript by Popovici et al., is an excellent retrospective study concerning the impact of the body mass index and prostate cancer survival. The authors analyzed the data obtained from a conspicuous number of patients over the past seven years. The results obtained indicate a strong correlation between being overweight and obesity and the possibility of developing advanced or metastatic prostate cancer.
Overall, this research is well-written, and the content of this manuscript is of major interest and I do not find any significant incorrectness. My following comments are of minor character:
- Line 21: Please correct the dates ( dots are not necessary).
- Line 38 and throughout: References should be formatted like this: “[1].” I mean the number of the reference(s) in a square bracket and then the endpoint of the sentence.
- Line 224: “Prostate cancer remains a global health burden and cases will continue to rise, but 222 pharmacological and technological advances are considerably improving and should 223 enable future precision and improved clinical outcomes”. References are needed here. Excellent studies have been conducted in recent years, the results of which are very promising. For example, you should cite at least these: doi: 10.1038/s41419-020-03256-5; doi: 10.1016/j.ejmech.2022.114435; doi: 10.1038/s41598-019-44452-4
- Line 291: Did the authors forget the acknowledgments?
- References: Please check the journal guidelines about the reference list style
Round 2
Reviewer 1 Report
The authors have addressed some points raised in previous review. The effort is to be commended.
It would be interesting to discuss what are their proposals among radical prostatectomy candidates. Since the findings highlighted that overweight and obese patients were more likely to have an advanced prostate cancer at time of diagnosis or radical treatment (intention-to-cure setting). A more aggressive surgical management in order to perform a debulking procedures in a well-selected cohort? Increasing the template of the lymph node dissection for example with intraoperative guidance (indocyanine-green, TC-labelled or PSMA-labelled)? Avoidance of positive surgical margins using intraoperative novel tools like confocal microscopy (doi: 10.1016/j.euf.2020.08.013; doi: 10.1016/j.eururo.2021.03.021)?
Author Response
As requested, we discussed our proposal for radical prostatectomy (298-316).
Note: OncoHelp Association Timisoara is only involved in the clinical management of cancer patients. To restrict a conflict between hospitals, we had to limit our data about radical prostatectomy, lymphadenectomy and histopathological exam. In Romania the urologist is involved only in the surgical management of prostate cancer and the oncologist is doing the rest.
